# Sulforaphane-Induced Cell Mitotic Delay and Inhibited Cell Proliferation via Regulating CDK5R1 Upregulation in Breast Cancer Cell Lines

**DOI:** 10.3390/biomedicines11040996

**Published:** 2023-03-23

**Authors:** Chao-Ming Hung, Tai-Hsin Tsai, Kuan-Ting Lee, Yi-Chiang Hsu

**Affiliations:** 1Department of General Surgery, E-Da Cancer Hospital, Kaohsiung 824, Taiwan; ed100647@edah.org.tw; 2School of Medicine, I-Shou University, Kaohsiung 824, Taiwan; 3Division of Neurosurgery, Department of Surgery, Kaohsiung Medical University Hospital, Kaohsiung 807, Taiwan; 4Department of Surgery, School of Medicine, College of Medicine, Kaohsiung Medical University, Kaohsiung 807, Taiwan; 5Graduate Institute of Medicine, College of Medicine, Kaohsiung Medical University, Kaohsiung 807, Taiwan; ayta860404@gmail.com; 6Division of Neurosurgery, Department of Surgery, Kaohsiung Municipal Ta-Tung Hospital, Kaohsiung 801, Taiwan

**Keywords:** breast adenocarcinoma, sulforaphane, cyclin dependent kinase 5 regulatory subunit 1, cell division cycle protein 2

## Abstract

Our research has revealed that sulforaphane (SFN) has chemopreventive properties and could be used in chemotherapy treatments. Further investigation is needed to understand the mechanisms behind sulforaphane’s (SFN) antitumor activity in breast adenocarcinoma, as observed in our studies. This research looked into the effects of SFN on mitosis delay and cell cycle progression in MDA-MB-231 and ZR-75-1 cells, two types of triple-negative breast cancer adenocarcinoma.The proliferation of the cancer cells after SFN exposure was evaluated using MTT assay, DNA content and cell cycle arrest induction by flow cytometry, and expressions of cdc25c, CDK1, cyclin B1 and CDK5R1 were assessed through qRT-PCR and Western blot analysis. SFN was found to inhibit the growth of cancer cells. The accumulation of G2/M-phase cells in SFN-treated cells was attributed to CDK5R1. The disruption of the CDC2/cyclin B1 complex suggested that SFN may have antitumor effects on established breast adenocarcinoma cells. Our findings suggest that, in addition to its chemopreventive properties, SFN could be used as an anticancer agent for breast cancer, as it was found to inhibit growth and induce apoptosis of cancer cells.

## 1. Background

Metastasis accounts for over 85% of breast cancer-related deaths in women, making it a primary cause of mortality [1]. Metastatic colonization is a complex process that involves multiple steps, including detachment of cancer cells from the primary tumor, invasion of the surrounding microenvironment, and intravasation into preexisting or newly formed blood vessels, and generate distant metastases [2]. The efficacy of conventional chemotherapy or chemoprevention for breast cancer is often unsatisfactory [3]. The severe and many side effects and limitations of conventional treatments have led to the widespread application of complementary and/or alternative medicines. Therefore, it is crucial to prioritize the development of novel compounds for molecular targeting in breast cancer research. While there are various treatment modalities available, traditional medicines with proven antitumor activity are preferred [4], though the use of naturally occurring compounds combined with chemotherapy might enhance drug sensitivity [5].

Natural products have been an unparalleled source of anti-cancer drugs in the modern age of drug discovery. Natural products play an important role in cancer therapy by modulating the cancer microenvironment and different signaling pathways. Therefore, it is necessary to study the current role of natural products in the discovery and development of anti-cancer drugs and determine the possibility of natural products as an important source of future therapeutic agents [6]. Here, we use sulforaphane (SFN) as a choice of anti-cancer selection of natural medicines for this study.

Sulforaphane (SFN), a potent preventive agent, is a dietary isothiocyanate found as a precursor glucosinolate in cruciferous vegetables [7]. Although the biological activity of SFN, including its antioxidation and anti-mutagenesis effects, has been widely reported [8,9], little is known about the mechanisms underlying its effects [10]. Here, we investigated whether SFN induces a reduction in breast cancer cell growth and the mechanism.

Cyclin-dependent kinases (Cdks) are a family of serine/threonine kinases that mainly function in the progression of cell cycles [11]. Cell cycle-associated Cdks, which are expressed in proliferating cells, require cyclin binding for activation, a process regulated by multiple phosphorylation [11]. Many studies have demonstrated that cyclin-dependent kinase 1 (CDK1) is associated with tumor aggressiveness due to its critical role in cell cycle progression and resistance to apoptosis induction [12]. Extensive research has been conducted on the activation mechanism of CDC2-cyclin B in chemotherapy. Specifically, CDC2 is phosphorylated at Thr-14 and Tyr-15 in the ATP-binding G-loop by either Wee1 or Myt1 kinase [13]. This phosphorylation at Tyr-15 keeps the CDC2/cyclin B complex in a state of inactivity until the beginning of M phase, at which point these sites are dephosphorylated by the phosphatase Cdc25 [14].

Cyclin-dependent kinase 5 (Cdk5) is a member of the serine/threonine and Cdk superfamily has been described as being crucially involved in regulating neuronal migration during brain development and neuron death [15]. Thus, phosphorylation at Tyr-15 appears to be important to Cdk5 functioning. CDK5 activity is controlled, in part, by binding to its regulatory subunit p35 and/or the p35-derived peptide p25, which contributes to the hyperactivation of CDK5 in certain disease states [16]. Cdk5 has been found to be first phosphorylated by c-Abl and then activated by binding p35 [17]. The role of CDK5 and possible effectors involved in the progression and metastatic spread of primary malignant melanoma have been studied in vitro and in vivo [18]. Binding to the activator component cyclin-dependent kinase 5 regulatory subunit 1 (CDK5R1) activates the Cdk5 kinase complex [19]. The Cdk5/Cdk5r1 complex is highly expressed in neuronal migration [20]. Furthermore, the deregulation of the Cdk5 kinase has been associated with the progression of Alzheimer’s disease, breast cancer, and prostate cancer [21,22]. CDK5R1 mediated proliferation is dependent on the Cdk5 complex kinase activity and CDK5R1 mediated proliferation correlates with the increased phosphorylation of pRb [23]. Apoptosis and cell cycle regulation are both potential targets for cancer therapy.

In this study, we accessed the effects of SFN on mitosis delay and cell cycle progression in the triple-negative breast cancer (TNBC) cell line MDA-MB-231 and ductal carcinoma cell line ZR-75-1 and investigated the possible underlying involvement of CDK5R1 in the achievement of these effects.

## 2. Methods

### 2.1. Materials

Sulforaphane (SFN), 3-(4,5-Dimethylthiazol-2-yl)-2,5-diphenyltetrazolium bromide (MTT) and dimethyl sulfoxide (DMSO) were purchased from Sigma (St. Louis, MO, USA). Dulbecco’s modified Eagle medium, fetal bovine serum (FBS), phosphate-buffered saline (PBS), sodium pyruvate, trypsin and antibiotics were purchased from Gibco, BRL (Grand Island, NY, USA). Polyvinylidene fluoride (PVDF) membranes were obtained from Millipore, and molecular weight markers were purchased from (Bio-Rad Laboratories Inc., Taiwan Branch, Taipei, Taiwan). All reagents and compounds were of analytical grade.

### 2.2. Cells

The MDA-MB-231 cells, are poorly differentiated human breast adenocarcinoma grade III cells. EGF receptors, TGF- receptors, WNT7B oncogene expression, and tumorigenicity (ATCC HTB-26) were maintained in 90% (*v*/*v*) L-15 with 2 mM L-glutamine and 1.5 g/L sodium bicarbonate and 10% (*v*/*v*) FBS. The cells were cultured at 37 °C in an atmosphere containing 100% air. We also studied ductal carcinoma ZR-75-1 (NCI-PBCF-CRL1500) cells. The medium used to grow ZR-75-1 cells contained RPMI 1640 and 10% (*v*/*v*) FBS. The ZR-75-1 cells were cultured at 37 °C in an air atmosphere containing 5% CO2.

### 2.3. Cell Proliferation Assay

The cells were seeded into a 96-well culture plate at 5000 cells/well and exposed to 0, 6.25, 12.5, or 25 ìM SFN for 24–72 h. The cells in each well were then treated with MTT (1 mg/mL) for at least 4 h. The reaction was stopped by adding DMSO, and the optical density at 540 nm (OD_540_) was measured using a multiwell plate reader. Background absorbance of the medium in the absence of cells was subtracted. All samples were assayed in triplicate and means calculated for each experiment. The results were expressed as a percentage of control, considered to be 100%. All assay results are expressed as mean ± SEM.

### 2.4. Apoptosis Measurement

The cells were first seeded in six-well culture plates (Orange Scientific, Braine-l’Alleud, Belgium). After treatment with SFN for 4 h, the cells were harvested and centrifuged. The cell pellet was then resuspended and incubated with 1× annexin-binding buffer [5 μL of annexin V-FITC (BD Pharmingen, Franklin Lakes, NJ, USA) and 1 μL of 100 μg/mL propidium iodide (PI) working solution] for 15 min at room temperature. After incubation, the stained cells were detected by a FACSCalibur flow cytometer (BD Pharmingen) and analyzed using WinMDI version 2.9 (BD Pharmingen).

### 2.5. Caspase-3 Activity Assay

Caspase activity was assessed using the FITC rabbit anti-active caspase-3 antibody (BD Pharmingen). The cells were treated with SFN for 1 day. Caspase activity was detected by the FACSCalibur flow cytometer and data were analyzed using WinMDI version 2.9.

### 2.6. Cell Cycle Analysis

We used the fluorescent nucleic acid dye PI to identify the proportion of cells in each of the three interphase stages of the cell cycle. The cells were treated with SFN for 24 h, harvested and fixed in 1 mL of cold 70% ethanol for at least 8 h at −20 °C. DNA was stained using a PI/RNaseA solution, and the DNA content was detected by a FACSCalibur flow cytometer. Data were analyzed using WinMDI version 2.9.

### 2.7. Evaluation of Mitochondrial Membrane Potential

The cells were seeded into 24-well plates (Orange Scientific, Braine-l’Alleud, Belgium). Following treatment with SFN for 6 h, we added 10 µg/mL JC-1 or 0.1 mg/mL Rhodamine 123 (Merck Ltd., Taipei, Taiwan) to the culture medium at 50 µL/well, which was then incubated at 37 °C for 20 min for mitochondrial staining. After being washed twice with warm PBS, the cells were fixed with 2% paraformaldehyde and detected by a FACS Calibur flow cytometer (JC-1). Data were analyzed using WinMDI version 2.9. Rhodamine 123 was detected using fluorescence microscopy (Olympus, Taichung, Taiwan).

### 2.8. Measurement of Intracellular Reactive Oxygen Species Generation

Intracellular reactive oxygen species (ROS) generation was measured using a DCFH-DA fluorescent dye (Molecular Probes, Eugene, OR, USA and Thermo Fisher Scientific, Waltham, MA, USA). The cells were cultured in six-well plates at a density of 1 × 10^4^ cells/well. The cells were treated with SFN, incubated with 10 µM DCFH-DA at 37 °C for 30 min, and then washed twice with PBS. For each experiment, the cells were analyzed for fluorescence by a FACSCalibur flow cytometer. Data were analyzed using WinMDI version 2.9.

### 2.9. Mitotic Index Analysis

Mitotic index was assessed based on MPM-2 (anti-phospho-Ser/Thr-Pro) expression. After treatment with SFN, cells were harvested and fixed in 70% ethanol overnight. The cells were then washed and suspended in 100 µL of IFA-Tx buffer (4% FCS, 150 nM NaCl, 10 nM HEPES, 0.1% sodium azide and 0.1% Triton X-100) with the MPM-2 antibody at room temperature for 1 h. The cells were then washed and resuspended in IFA-Tx buffer with a rabbit antimouse FITC-conjugated secondary antibody (1:50 dilution; Serotec, Oxford, UK) for 1 h at room temperature in the dark. Finally, the cells were washed and resuspended in 500 µL of PBS with 20 µg/mL PI (Sigma-Aldrich, St. Louis, MO, USA) for 30 min in the dark. MPM-2 expression was measured by FACSCalibur flow cytometer. Data were analyzed using WinMDI version 2.9.

### 2.10. Confocal Microscopy

Confocal microscopy was performed as described previously. Briefly, 2 × 10^6^ cells were treated with SFN for 16 h and fixed on coverslips. After treatment, they were incubated with 1 µg/mL primary MPM-2 antibody (Upstate Cell Signaling Solutions, Millipore, Watford, UK) for 30 min and washed with PBS. The cells were incubated with a rabbit anti-mouse FITC-conjugated secondary antibody (1:50 dilution; Serotec, Oxford, UK) for 1 h at room temperature in darkness and then washed with PBS. These cells were then mounted onto microscope slides using a mounting medium containing DAPI.

### 2.11. Western Blot Analysis

Proteins (50–75 µg) were separated through 10–12% SDS-PAGE and transferred to PVDF membranes (Merck Ltd. Taiwan, Taipei, Taiwan). The membranes were blocked with a blocking buffer (Odyssey LI-COR, Lincoln, NE, USA) overnight and incubated with anti-actin (Sigma-Aldrich) and anti-p-cell division cycle 25 (cdc25c) (sc-12354), anti-cdc25c (sc-13138), anti-CDC2 (p34; sc-747), anti-p-CDC2 (sc-7989), anti-cyclin B1 (sc-752) and CDK5R1 antibodies for 90–120 min. The blots were then washed and incubated with a secondary antibody at a 1:20,000 dilution for 30–45 min. Next, the antigens were visualized on a near-infrared imaging system (Odyssey). Data were analyzed using the software Odyssey 2.1 and a chemiluminescence detection kit (ECL; Amersham Corp., Arlington Heights, IL, USA). Data were analyzed using Odyssey version 2.1 software.

### 2.12. Co-Immunoprecipitation (Co-IP)

Co-IP is an effective means of quantifying protein–protein interactions in cells. Briefly, after incubation at room temperature overnight, 500 mg of cellular proteins were labeled using anti-CDC2 (p34; sc-747). The protein–antibody immunoprecipitates were collected using protein A/G plus-agarose beads (SC-2003 Santa Cruz BioTechnology, Dallas, TX, USA). Following the final wash, the samples were boiled and centrifuged to transform the agarose beads into pellets. Finally, the supernatant containing the eluted proteins was separated from the beads and analyzed by western blot. Antigens were visualized using a chemiluminescence detection kit (ECL; Amersham Corp., Arlington Heights, IL, USA), and data were analyzed using Odyssey version 2.1 software.

### 2.13. Gene Expression Profiling (GEP)

Briefly, the cells untreated or treated with SFN for 4 h were harvested and total RNA was isolated utilizing an RNasey kit (Qiagen, Hilden, Germany) following manufacturer’s directions. Total RNA was sent to Welgene Company (Taipei, Taiwan) for whole human genome SurePrint G3 array GEP analysis (Agilent Technologies, Santa Clara, CA, USA).

### 2.14. Quantitative Real-Time PCR

Quantitative real-time PCR (qRT-PCR) was performed using approximately 200 ng of SYBR Green PCR MasterMix in an ABI 7300 system (Applied Biosystems, Foster City, CA, USA). Forty PCR cycles were conducted at the following temperatures and durations: 95 °C for 120 s, 60 °C for 30 s, and 72 °C for 30 s. Sample cells from three plates were run in duplicate and the threshold suggested by the software was adopted for Ct calculations. To normalize readings, we used Ct values obtained at 18 s as internal controls for each run and were thus able to calculate a delta Ct value for each gene. All procedures used to perform qRT-PCR were in accordance with the manufacturer’s instructions.

### 2.15. Small-Interfering RNA

The specific small-interfering RNA of CDK5R1 (GenePharma, Shanghai, China) and the Lipofectamine RNAiMAX gene transfection system were purchased from Invitrogen (Thermo Fisher Scientific, Waltham, MA USA). Duplex siRNA was re-suspended in RNase-free buffer (10 mM Tris-HCl, pH 8.0, 20 mM NaCl, 1 mM EDTA). The transfection protocol is based on procedures described in the Lipofectamine RNAiMAX’s manual.

### 2.16. Statistical Analysis

All data are reported as the mean (±SEM) of at least three separate experiments. The *t* test or one-way ANOVA with a post hoc test was used for statistical analysis. A *p* < 0.05 was considered significant. All statistical operations were performed using SPSS 24.0 software.

## 3. Results

### 3.1. Non-SFN-Induced Apoptosis/Necrosis of Breast Cancer Cell Lines

This study hypothesized that SFN might inhibit the survival of MDA-MB-231 and ZR-75-1 cells and thereby prevent their proliferation. To find out, we treated MDA-MB-231 and ZR-75-1 cells with increasing doses of SFN (0, 6.25, 12.5, and 25 ìM) for 24–72 h and then measured the proliferation of SFN-treated cancer cells by MTT assay (Figure 1A). The higher the SFN does, the greater the decrease in survival and proliferation of the MDA-MB-231 and ZR-75-1 cells. Microscopic examination further revealed morphological changes in the cells following exposure to SFN (12.5 ìM) for 6–24 h. SFN also induced cancer cell death, indicated by the dead cell suspension in the medium.

To investigate the role of SFN in inducing apoptosis/necrosis in breast cancer cells, we utilized annexin V-FITC and PI staining to observe the formation of apoptotic cells following a 4-h exposure to SFN. We evaluated the percentage of apoptotic cells using flow cytometry. Appendix A shows a dot-plot of annexin V-FITC fluorescence versus PI fluorescence, which reveals that SFN-treated cells exhibited a non-significant increase in the percentage of apoptotic cells when compared with untreated (control) cells (Appendix AA). Additionally, there were no significant changes in the percentage of SFN-treated breast cancer cells undergoing necrosis, apoptosis, or caspase-3 activity (Appendix A).

### 3.2. Assessment of Changes in Mitochondrial Membrane Potential

The loss of mitochondrial membrane potential (ΔΨm) is an early cellular metabolism event that occurs simultaneously with caspase activation and is a hallmark of apoptosis. In non-apoptotic cells, JC-1 is present in the cytosol as a green monomer, but it accumulates as red aggregates in the mitochondria. However, in apoptotic and necrotic cells, JC-1 is present in the monomeric form and stains the cytosol green. To investigate the effect of SFN on ΔΨm in breast cancer cell lines, we used rhodamine 123 as a dye to assess the loss of ΔΨm in SFN-treated cancer cell lines. Our results (Appendix A) showed that SFN treatment led to a reduction in the ΔΨm of the breast cancer cells. Furthermore, typical FL-1/FL-2 dot plots for apoptotic and non-apoptotic MDA-MB-231 and ZR-75-1 cells stained by JC-1 (Appendix A) indicated that SFN treatment resulted in a nonsignificant reduction in ΔΨm of the MDA-MB-231 and ZR-75-1 cells (Appendix A). These findings suggest that SFN may mediate the survival of breast cancer cell lines through pathways other than those related to apoptosis/necrosis. Therefore, we hypothesized that the proliferation of these cells was inhibited by mechanisms other than those of apoptosis/necrosis. These results are summarized in Appendix A. Based on the results summarized in Appendix A, it can be hypothesized that SFN may mediate breast cancer cell line survival by inhibiting pathways other than those of apoptosis/necrosis.

### 3.3. SFN Reduced ROS Accumulation in Cells

Chemotherapy induces cell death in various tumor types, in part by promoting intracellular ROS generation. To determine whether ROS generation was associated with the SFN-induced arrest of growth in MDA-MB-231 and ZR-75-1 cells, we measured ROS in these cell lines following the administration of varying doses of SFN. Flow cytometry was used to examine the fluorescence intensity of DCHF-DA-incubated cells. The representative fluorescence patterns of the control and SFN treatment groups revealed a significant decrease in intracellular ROS levels. We found that SFN reduced ROS accumulation in MDA-MB-231 and ZR-75-1 cells, suggesting that SFN-induced growth arrest in MDA-MB-231 cells was caused by the inhibition of intracellular ROS accumulation leading to cell growth inhibition.

We studied cell cycle distribution of SFN-treated cells. Cells were first exposed to SFN for 24 h and then processed and analyzed by flow cytometry. As shown in Figure 1B, exposure to SFN resulted in an increase in the number of cells in the G_2_/M phase, suggesting that the MDA-MB-231 and ZR-75-1 cells had undergone a delay in mitosis. These results indicated that SFN increased the number of cell populations in the G_2_/M phase while simultaneously reducing the number of cells in the G_1_ phase (Figure 1C).

### 3.4. Effects of SFN on the Mitotic Index

In order to distinguish between G2 arrest and mitotic arrest, we employed the marker MPM-2 (anti-phospho-Ser/Thr-Pro)-FITC (green in Figure 2A). This antibody is capable of recognizing proteins with epitopes that are exclusively phosphorylated during mitosis, specifically from the early prophase to metaphase [24]. MPM-2 is often used to indicate disturbances in mitosis. To validate the effectiveness of our experimental setup, we treated separate groups of cells with nocodazole (15 µg/mL), a substance known to induce metaphase arrest, for 24 h [25]. This treatment resulted in the synchronization of entire cell populations in the G2/M phase and increased labeling of MPM-2, as shown in (Figure 2B).

According to the statement, the MPM-2 levels in MDA-MB-231 and ZR-75-1 cells treated with SFN were higher than those in the control group (Figure 2C). This suggests that SFN treatment caused an increase in phosphorylation of proteins recognized by MPM-2, which is consistent with mitotic disturbance. It is possible that the strong effects of staining with MPM-2 in the group treated with 25 μM SFN were due to the presence of cells in various stages of mitosis, some of which could not be identified using this early prophase marker. In addition, the higher concentration of SFN used in this experiment may have contributed to a greater level of mitotic disturbance, resulting in an increase in MPM-2 labeling. Further studies may be needed to fully understand the effects of SFN on mitotic progression. Some cells in the G2/M phase may have not been detected by MPM-2 staining, which could lead to an underestimation of the level of mitotic disturbance indicated by the increased MPM-2 staining. Taken together, it is shown that MDA-MB-231 and ZR-75-1 cells were treated with SFN to increase MPM-2 levels, which resulted in G2 arrest from mitotic arrest.

### 3.5. G_2_/M-Phase Cell Cycle Arrest by SFN in the MDA-MB-231 Cells through CDC2 and Cyclin B1 Disassociation

Figure 3A,B present the results of qRT-PCR and Western blot analysis for cellular proteins from the MDA-MB-231 and ZR-75-1 cells treated with SFN. Gene expression analysis showed that cyclin B1 was upregulated, but CDC2 levels did not significantly change following incubation with SFN in MDA-MB-231 and ZR-75-1 cells (Figure 3A). In this test, CDC2 and cyclin B1 gene expression levels were quantified by measuring relative intensities. We observed no significant change in CDC2 levels following incubation with SFN (Figure 3B,C). Furthermore, the levels of cyclin B1 were significantly increased in those cells incubated with SFN at concentrations of 6.25 to 25 ìM (Figure 3C). We also quantified the activity of the cyclin B1/CDC2 complex (crucial for G_2_–M transition during the cell cycle) using a Co-IP test (Figure 3D). We found an increase in the number of the MDA-MB-231 and ZR-75-1 cells in the G_2_/M phase attributed to the disassociation of CDC2 and the cyclin B1 complex following incubation with SFN. The disassociation of CDC2/cyclin B1 complex in MDA-MB-231 cells by SFN may occur through the phosphorylation of CDC2 (Figure 3B, Appendix A) and reduce the CDC2 activity in breast cancer cells (Figure 4A).

### 3.6. Effects of SFN on CDC25C in MDA-MB-231 and ZR-75-1 Cells

Figure 3B illustrates p-CDC25C and CDC25C expression and immunoblotting results for cellular proteins in SFN-treated breast cancer cells. Protein expression was quantified by measuring relative band intensities (Appendix A–C). Western blot analysis revealed a decrease in the CDC25C: p-CDC25C ratio in MDA-MB-231 cells, but an increase in ZR-75-1 cells after incubation with SFN (Figure 4B). These data suggest that SFN is involved in CDC25C regulation in MDA-MB-231 and ZR-75-1 cells.

### 3.7. Gene Expression Profile of Cells following Exposure to SFN

We used SurePrint G3 Human Gene Expression Microarrays to study the genome-wide gene expression profiles of MDA-MB-231 cells exposed to either the vehicle (DMSO) or SFN (25 ìM) for four hours. The experiments were performed independently three times to enable comparative analysis between the MDA-MB-231 cells. Principal component analysis (PCA) of microarray data derived from SFN-treated cells and DMSO-treated cells revealed spatially separated planes, suggesting that SFN had a far greater impact on the GEP than could be reasonably attributed to technical error. We divided the expression levels in the SFN-treated cells by those in the vehicle-treated group to analyze up- and down-regulation, considering changes greater than 2-fold to be substantial upregulation and changes smaller than 0.5-fold to be downregulation. To identify molecular networks of biological significance for these genes, we used distinct pathway bioinformatics analysis tools as well as the comprehensive knowledgebase of the Kyoto Encyclopedia of Genes and Genomes (KEGG) (www.kegg.jp, (accessed on 20 July 2021)). This enabled us to identify the KEGG pathway (Appendix A). We found that SFN increased the levels of CDK5R1 (Figure 5A,B) but not CDK5 in MDA-MB-231 cells.

### 3.8. Down-Regulation of the CDK5R1 in MDA-MB-231 and ZR-75-1 Cells by Silencing CDK5R1 Was Reversed by SFN Induced G_2_/M Arrest

To confirm the relationship between CDK5R1 expression and cell cycle arrest in MDA-MB-231 cells, we modulated cellular CDK5R1 levels using Lipofectamine RNAiMAX gene delivery techniques. We then investigated the influence of cellular CDK5R1 levels on anti-tumorigenic behavior in MDA-MB-231 and ZR-75-1 cells. Protein analysis revealed a significant decrease in the levels of the CDK5R1 after CDK5R1 was silenced (Figure 6A). qPCR analysis also revealed higher levels of CDK5R1 mRNA in the cells treated with SFN (Figure 5A). In addition, silencing CDK5R1 significantly repressed SFN-induced cell cycle G2/M arrest in MDA-MB-231 cells (Figure 6B,C). These findings suggest that the CDK5R1 level regulated the tumorigenicity of MDA-MB-231 cells via SFN.

Considered together, our findings indicate that common molecular pathways are involved in inducing mitosis delay. Findings from qPCR analysis were further validated by a microarray analysis, indicating substantial CDK5R1 upregulation (Figure 5A) as well as the notable upregulation of p-CDC25C, p-CDC2 and the cyclin B1, CDC2 dissociation in MDA-MB-231 and ZR-75-1 cells following exposure to SFN. These results indicate that SFN may delay cancer cell growth in the G_2_/M phase via the dissociation of the cyclin B1/CDC2 complex and up-regulation of CDK5R1 proteins.

## 4. Discussion

Our experimental results offer evidence supporting the idea that SFN can induce irreversible growth arrest in MDA-MB-231 and ZR-75-1 cells. Mechanistic analysis revealed that SFN’s ability to inhibit cell proliferation and cause delays in mitosis are strongly associated with its accumulation within these cell lines.

In our observations, SFN treatment of cancer cells led to the decrease of cyclin B1/CDC2 complex association and phosphorylation of CDC25C. Furthermore, CDC25C was found to interact with the CDC2/cyclin B1 complex, which is responsible for regulating both the G2 and M phases of the cell cycle [26]. The CDC25 protein family, which are highly conserved dual-specificity phosphatases, play a crucial role in activating CDK complexes and their corresponding cyclins, thereby regulating the progression of the cell cycle [27]. During cell division, CDC25C dephosphorylates CDC2, activating the CDC2/cyclin B complex, which results in entry into mitosis. In addition, the CDC2/cyclin B complex phosphorylates CDC25C, enhancing its phosphatase activity and leading to an irreversible auto-amplification loop into mitosis [28]. It should be noted that several studies have shown that the inactivation or degradation of CDC25C promotes the development of cancer by accelerating mitosis [29,30,31].

SFN-induced delay in mitosis does not necessarily occur through the downregulation of cyclin B1 and CDC2 gene expression (Figure 3B,C), but rather through the upregulation of CDK5R1 (Figure 5 and Figure 6) and CDC25C (Figure 4) phosphorylation in the MDA-MB-231 and ZR-75-1 cells.

Our experimental evidence thus indicates that SFN may irreversibly arrest the growth of cancer cells and that the inhibition of proliferation and induction of cell cycle arrest are both highly dependent on the SFN accumulation in cancer cells.

## 5. Conclusions

In conclusion, this is the first study to demonstrate that SFN is an effective inhibitor of MDA-MB-231 and ZR-75-1 cells, an effect achieved via its effect on CDK5R1. The upregulation of CDK5R1 delayed mitosis. However, SFN was not observed to have any significant inhibitory effects on human normal cells [32]. Our findings suggest that SFN might be used as an anticancer agent for breast cancer.

## Figures and Tables

**Figure 1 biomedicines-11-00996-f001:**
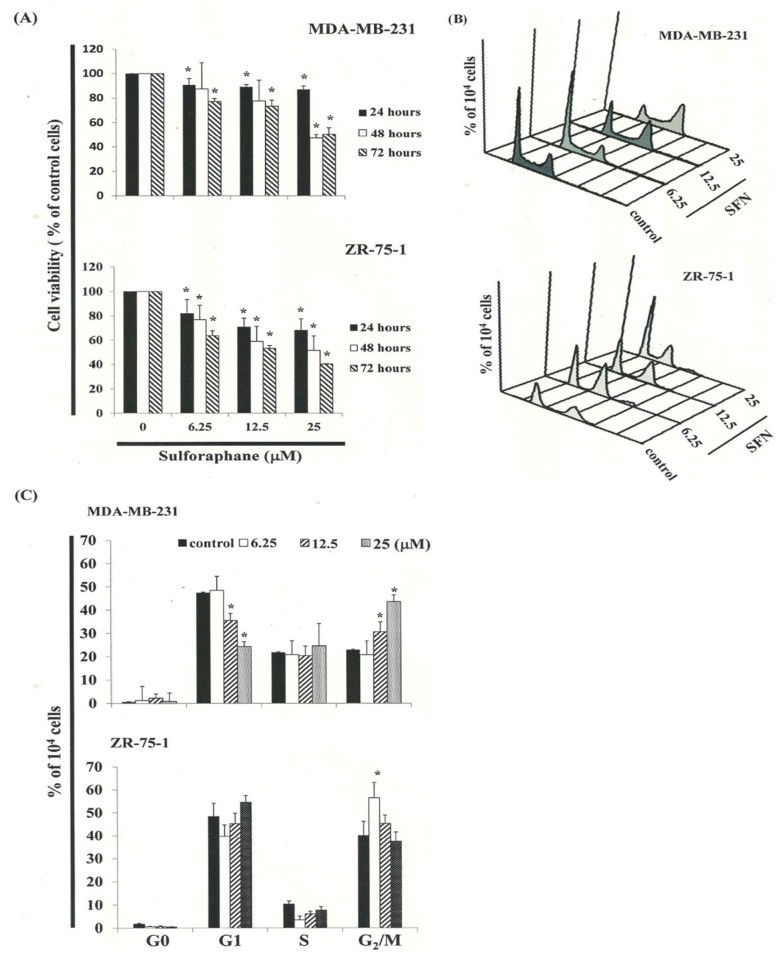
SFN mediates the survival of MDA-MB-231 and ZR-75-1 cells by inhibiting proliferation: (**A**) cells were treated with increasing doses of SFN (0, 6.25, 12.5, and 25 μM) for 24–72 h in vitro. The survival of SFN-treated cancer cells was measured using MTT assay. Results are expressed as a percentage of the control, considered to be 100%. All data are reported as the mean (±SEM) of at least three separate experiments. Statistical analysis was performed using the *t* test, with differences between the treatment and control groups (0 μM SFN) considered significant at *p* < 0.05, delineated by *. Effect of SFN on cell cycle progression and distribution in MDA-MB-231 and ZR-75-1 cells: (**B**) cell cycle analysis of the cancer cells after being cultured with SFN for 24 h. (**C**) SFN induced an increase in G_2_/M-phase cell percentage (%). Cells were stained with propidium iodide to analyze DNA content, which was then quantified by flow cytometry. * in each group of bars indicates that the number of G_2_/M cells in the SFN treatment group was significantly higher than that of the control group (*p* < 0.05).

**Figure 2 biomedicines-11-00996-f002:**
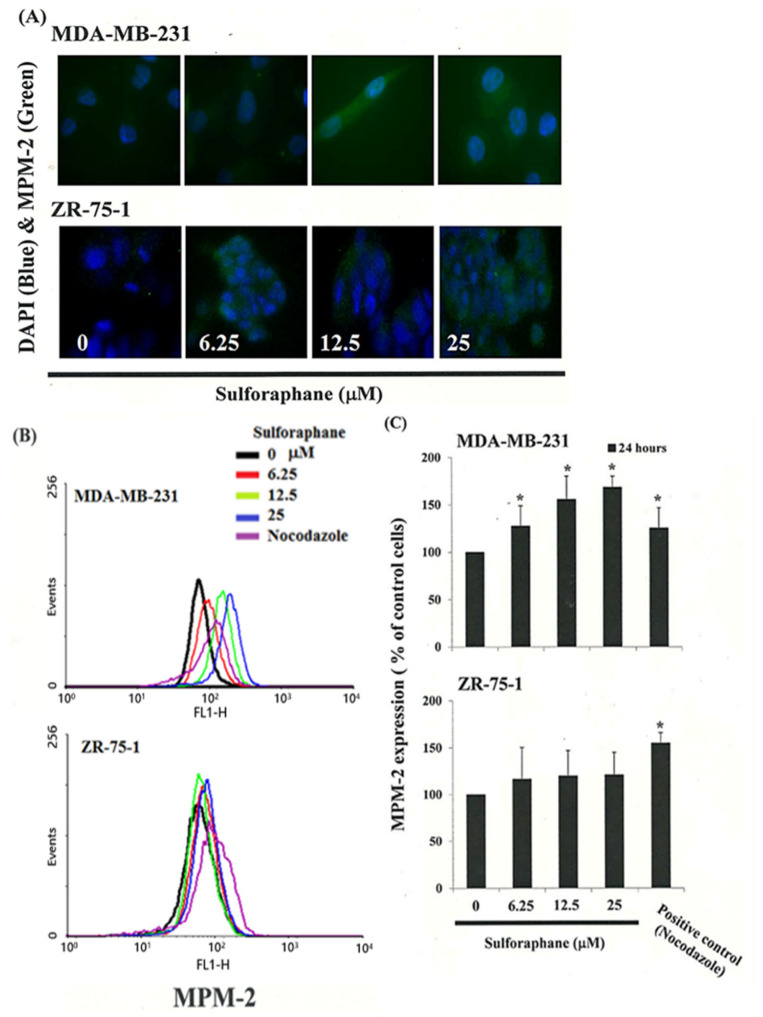
Diminished MPM-2 activity in SFN-treated MDA-MB-231 and ZR-75-1 cells. The MPM-2 activity of the cells was examined through confocal microscopy and flow cytometry 24 h after SFN stimulation. (**A**) The cells were stained for MPM-2 (green). DAPI (blue) stained nuclei with the active form of MPM-2. (**B**) Quantification of MPM-2 expression (gated cells) was performed by flow cytometry following treatment with SFN for 24 h. (**C**) SFN increased the level of MPM-2 in those cells. As a positive control, separate groups of cells were treated for 24 h with Nocodazole (15 µg/mL), an antifungal agent that induces metaphase arrest. * in each group of bars indicates that the difference resulting from treatment with 0 ìM SFN was significant at *p* < 0.05.

**Figure 3 biomedicines-11-00996-f003:**
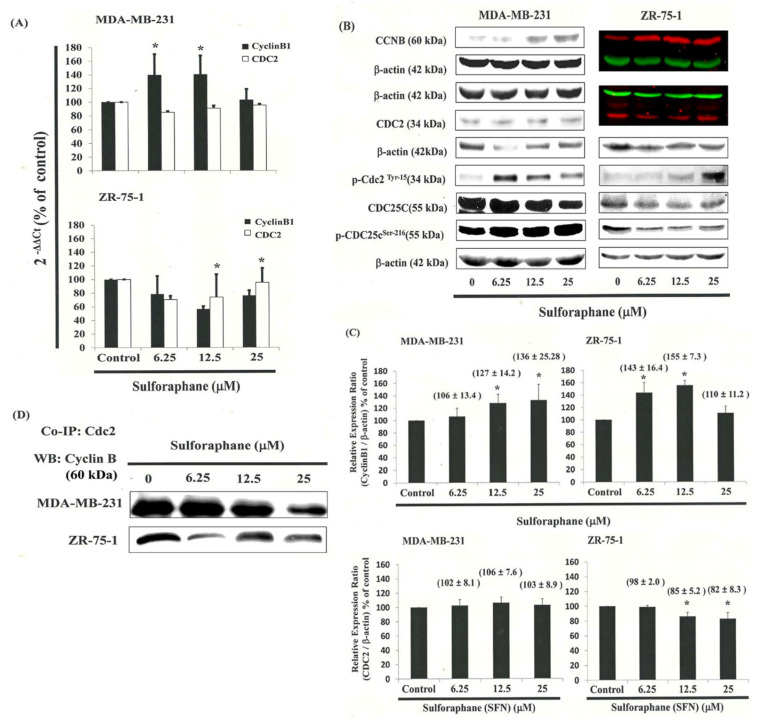
SFN repressed CDC25C activity and enhanced CDC2/cyclinB1 disassociation in MDA-MB-231 and ZR-75-1 cells. Cells were treated with SFN for 24 h, the gene and protein expression was subsequently detected using (**A**) q-RT-PCR and (**B**) Western blot analysis: (**C**) representative blots from three independent experiments. Quantification of band intensities. (**D**) Co-IP of CDC2 and cyclinB1 in breast cancer cell lines treated with SFN. * in each group of bars indicates that the difference resulting from treatment with 0 ìM SFN was significant at *p* < 0.05. * in each group of bars indicates that the difference resulting from treatment with 0 ìM SFN was significant at *p* < 0.05. All data are reported as the mean (±SEM) of at least three separate experiments.

**Figure 4 biomedicines-11-00996-f004:**
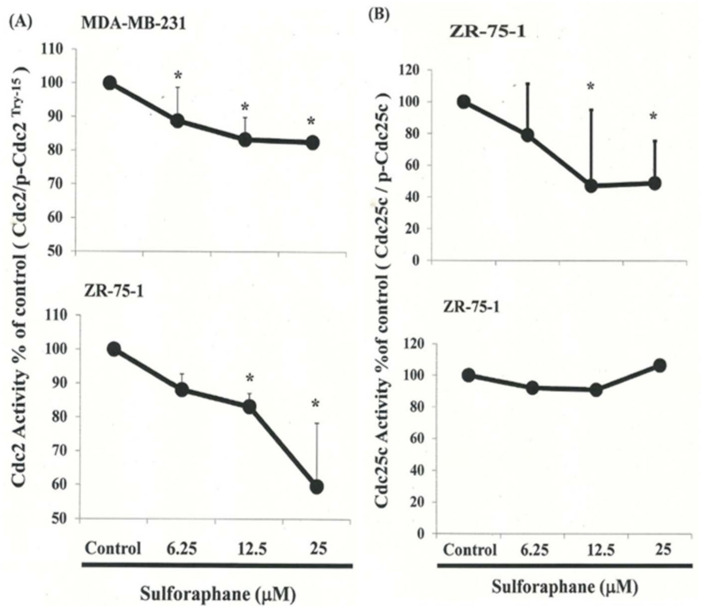
SFN repressed CDC2, CDC25C activity in MDA-MB-231 and ZR-75-1 cells. Cells were treated with SFN for 24 h, protein expression was subsequently detected by Western blot analysis, representative blots from three independent experiments. Quantification of band intensities. (**A**) CDC2 activity; (**B**) cdc25c activity. * in each group of bars indicates that the difference resulting from treatment with 0 ìM SFN was significant at *p* < 0.05. All data are reported as the mean (±SEM) of at least three separate experiments.

**Figure 5 biomedicines-11-00996-f005:**
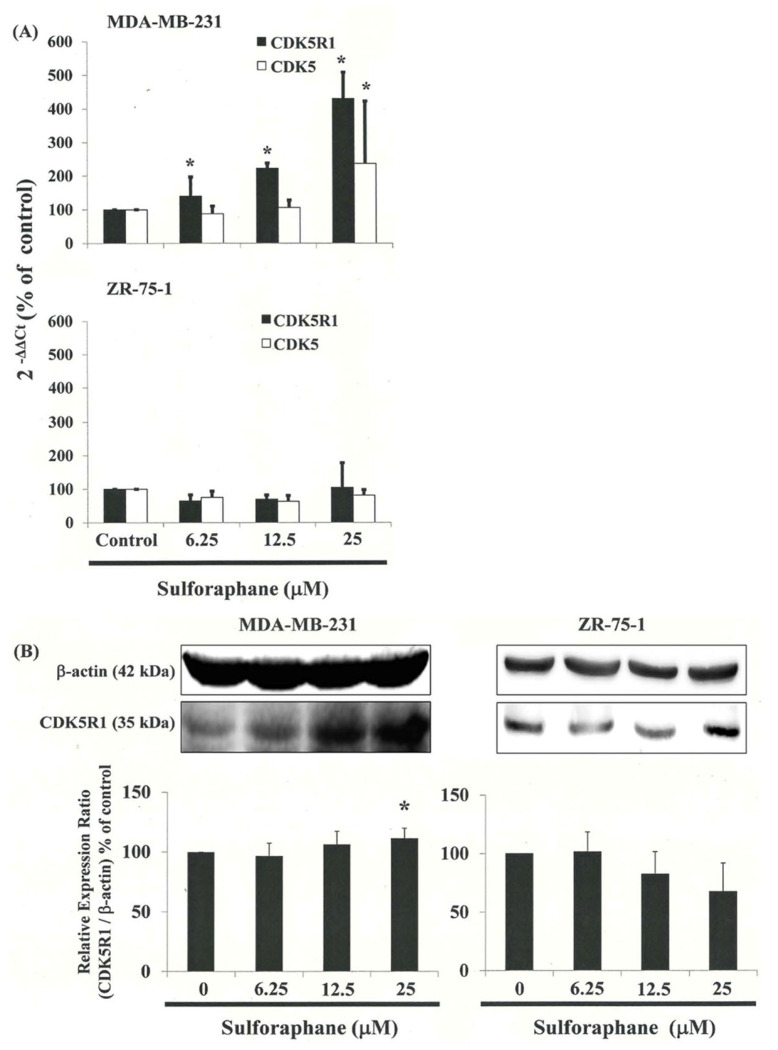
CDK5R1 and CDk5 gene expression in MDA-MB-231 and ZR-75-1 cells following exposure to SFN: (**A**) qPCR and (**B**) Western blotting analysis of CDK5R1 and CDk5 gene expression standardized against the levels of GAPDH or β-actin in cancer cell lines exposed for 4 h to DMSO (SFN 0 M control) or SFN. * in each group of bars indicates that the difference resulting from treatment with 0 ìM SFN was significant at *p* < 0.05. All data are reported as the mean (±SEM) of at least three separate experiments.

**Figure 6 biomedicines-11-00996-f006:**
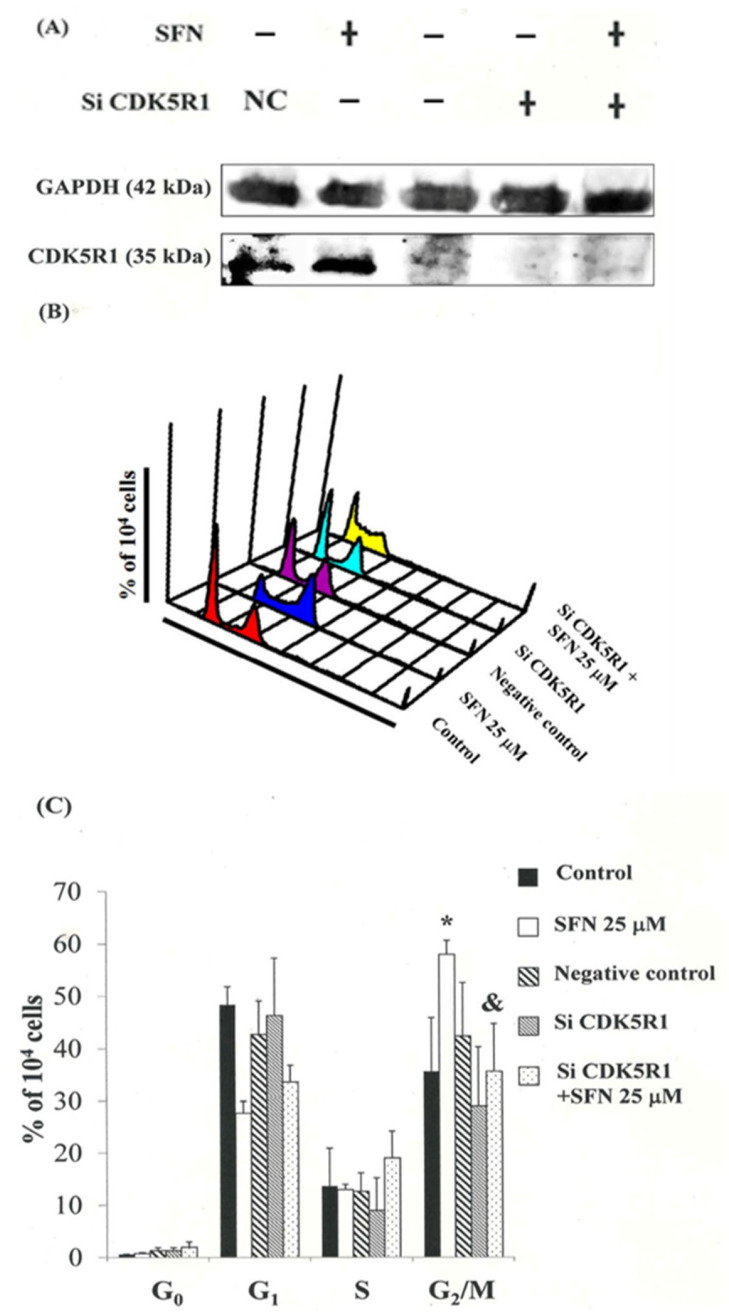
Re-enter cell cycle arrested by SFN in MDA-MB-231 and ZR-75-1 cells via inhibition of CDK5R1. (**A**) CDK5R1 in MDA-MB-231 and ZR-75-1 cells treated with SFN and/or CDK5R1 SiRNA. (**B**) Delay in mitosis in SFN-treated and/or reenter cell cycle CDKR1 SiRNA cells. (**C**) Cell cycle-phase percentage (%). Cells underwent staining with propidium iodide to analyze DNA content, which was quantified by flow cytometry. * in each group of bars indicates that the number of G_2_/M cells in the SFN treatment group was significantly higher than that of the control group (*p* < 0.05). All data are reported as the mean (±SEM) of at least three separate experiments.

## Data Availability

Not applicable.

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
