# Peer review of "Sulforaphane-Induced Cell Mitotic Delay and Inhibited Cell Proliferation via Regulating CDK5R1 Upregulation in Breast Cancer Cell Lines"

_biomedicines, 2023, doi:10.3390/biomedicines11040996_

Round 1

Reviewer 1 Report

I read with great interest the article „Sulforaphane-induced cell mitotic delay and reactive oxygen

species-related signaling via regulating CDK5R1 in Breast cancer cell lines” by Chao-Ming Hung, Tai-Hsin Tsai, Kuan-Ting Lee and Yi-Chiang Hsu.

In my opinion, the article is well-written, structured and the material is well-chosen. Results correctly presented and visualized. Discussion was conducted well, relevant works were cited.
The article presents the current knowledge on the important topic of the chemo-preventive effects of sulforaphane 18 (SFN) and its potential use in chemotherapy. From a scientific point of view, the work is fine. Contains important insights. Apart from minor linguistic corrections, which the authors will surely improve, I believe that the article is suitable for printing in its current form.

Author Response

Department of Surgery

Kaohsiung Municipal Ta-Tung Hospital

Kaohsiung, Taiwan

March 3, 2023

Dear Editor:

Thank you for giving me the opportunity to submit a revised draft of our manuscript which titled: Sulforaphane-induced cell mitotic delay and-inhibited cell proliferation via regulating CDK5R1 upregulation in Breast cancer cell lines. We appreciate the time and effort that you and the reviewers have dedicated to providing your valuable feedback on our manuscript. We are grateful to the reviewers for their insightful comments on this paper. We have been able to incorporate changes to reflect most of the suggestions provided by the reviewers. We have highlighted the modification within the manuscript and point-by-point response to the reviewers’ comments and concerns in this revision.

Comments from Reviewer 1

I read with great interest the article „Sulforaphane-induced cell mitotic delay and reactive oxygen species-related signaling via regulating CDK5R1 in Breast cancer cell lines” by Chao-Ming Hung, Tai-Hsin Tsai, Kuan-Ting Lee and Yi-Chiang Hsu.

In my opinion, the article is well-written, structured and the material is well-chosen. Results correctly presented and visualized. Discussion was conducted well, relevant works were cited. The article presents the current knowledge on the important topic of the chemo-preventive effects of sulforaphane 18 (SFN) and its potential use in chemotherapy. From a scientific point of view, the work is fine. Contains important insights. Apart from minor linguistic corrections, which the authors will surely improve, I believe that the article is suitable for printing in its current form.

Response to the Reviewer 1:

Thank you for your comment on this article. It would have been interesting to explore this aspect. Our findings suggest that SFN inhibited growth and induced apoptosis of cancer cells, suggesting that in addition to its known ability to prevent breast cancer, SFN could potential be used as an anticancer agent for breast cancer. We will make minor language corrections and revisions.

Reviewer 2 Report

This manuscript has demonstrated that sulforaphane-induced cell mitotic delay in breast cancer cell lines is mediated through CDK5R1. However, the discussion and results sections are poorly described and could be improved.

1. Although the title said “reactive oxygen-related signaling via regulating CDK5R1”, the authors did not provide data on the relationship between ROS and CDK5R1. If the title remains, please provide the data.

2. In line 61, please describe in which molecule “Thy-15” is contained.

3. In line 68, are CDK5R1 and the aforementioned p35 the same molecule? Please explain a little more about the CDK5R1.

4. In line 215-220, why did the authors analyze cell death 4 hours after SFN stimulation? Are microscopic dead cells used for these analyses?

5. In line 236-237, although the authors hypothesized that cell proliferation was inhibited by pathways other than apoptosis/necrosis, please describe the cause of cell death in line 211-212 in the discussion section.

6. In line 243-247, although the authors described that SFN-induced growth arrest occurred by mechanisms other than intracellular ROS accumulation, the results in Fig. S1C appear to cause ROS accumulation due to SFN. Please describe the basis for this claim in the discussion section.

7. In line 277-279, please describe the results of the ZR-75-1 cells.

8. In line 300-301, please describe the results of qPCR analyses.

9. In line 313-315, please describe the results of the ZR-75-1 cells.

10. In line 405-406, in order to apply it as anticancer agent, it is necessary to consider the effect on normal cells. Please cite some papers and describe the effect on normal cells in the discussion section.

11. Some experiments have shown different results for the two cell lines. Please discuss in the discussion section the pathway through which SFN-induced growth arrest of ZR-75-1 cells occurs.

Author Response

Department of Surgery

Kaohsiung Municipal Ta-Tung Hospital

Kaohsiung, Taiwan

March 3, 2023

Dear Editor:

Thank you for giving me the opportunity to submit a revised draft of our manuscript which titled: Sulforaphane-induced cell mitotic delay and-inhibited cell proliferation via regulating CDK5R1 upregulation in Breast cancer cell lines. We appreciate the time and effort that you and the reviewers have dedicated to providing your valuable feedback on our manuscript. We are grateful to the reviewers for their insightful comments on this paper. We have been able to incorporate changes to reflect most of the suggestions provided by the reviewers. We have highlighted the modification within the manuscript and point-by-point response to the reviewers’ comments and concerns in this revision.

Comments from Reviewer 2:

This manuscript has demonstrated that sulforaphane-induced cell mitotic delay in breast cancer cell lines is mediated through CDK5R1. However, the discussion and results sections are poorly described and could be improved.

Comment 1: Although the title said “reactive oxygen-related signaling via regulating CDK5R1”, the authors did not provide data on the relationship between ROS and CDK5R1. If the title remains, please provide the data.

Response to the comment 1:

Thanks for pointing this out. We agree with this comment. We decided to modify the title: Sulforaphane-induced cell mitotic delay and-inhibited cell proliferation via regulating CDK5R1 upregulation in Breast cancer cell lines.

(As Title; Line 2 -Line 4)

Comment 2: In line 61, please describe in which molecule “Thy-15” is contained.

Response to the comment 2:

Thanks to the reviewers for pointing out this problem. We describe that phosphorylation of Tyr-15 occurs only on monomeric Cdk5, and co-expression of activators p35/p25, p39, or cyclin I inhibit phosphorylation. Activation of Cdk5-p35 by Tyr-15 phosphorylation may play a role in neurite and spine retraction, dendritic outgrowth, and neuronal death. This is important for the function of Cdk5.

(As Background; Line 61)

References:

Kobayashi H, Saito T, Sato K, et al. Phosphorylation of cyclin-dependent kinase 5 (Cdk5) at Tyr-15 is inhibited by Cdk5 activators and does not contribute to the activation of Cdk5. J Biol Chem. 2014;289(28):19627-19636. doi:10.1074/jbc.M113.501148

Comment 3: In line 68, are CDK5R1 and the aforementioned p35 the same molecule? Please explain a little more about the CDK5R1.

Response to the comment 3:

Thanks to the reviewers for pointing out this problem. The protein encoded by this gene (p35) is a neuron-specific activator of cyclin-dependent kinase 5 (CDK5); the activation of CDK5 is required for proper development of the central nervous system. Furthermore, Cdk5 has a similar structure to other cyclin-dependent kinases, its activators are highly specific (CDK5R1 and CDK5R2). CDK5R1 activates the Cdk5 kinase complex. The Cdk5/Cdk5r1 complex is highly expressed in neuronal migration.

(As Background; Lines 60-69)

Comment 4: In line 215-220, why did the authors analyze cell death 4 hours after SFN stimulation? Are microscopic dead cells used for these analyses?

Response to the comment 4:

Thanks to the reviewers for pointing out this problem. We agree with this comment. In normal healthy cells, the phospholipid phosphatidylserine (PS) is located on the cytoplasmic surface of the plasma membrane. Due to the apoptosis, the plasma membrane undergoes structural changes that include translocation of PS from the inner to the outer leaflet (extracellular side) of the plasma membrane. Therefore, we evaluate SFN-stimulated cell death for 4 hours whether PS membrane eversion due to apoptosis was an early response. We observed morphological changes in the cells following exposure to SFN (12.5 μM) for 6–24 h by microscopic confirmed MTT assay. (As Result; Lines 215-220)

Comment 5: In line 236-237, although the authors hypothesized that cell proliferation was inhibited by pathways other than apoptosis/necrosis, please describe the cause of cell death in line 211-212 in the discussion section.

Response to the comment 5:

Thanks to the reviewers for pointing out this problem. We agree with this comment. When MDA-MB-231 and ZR-75-1 were exposure to SFN (0, 6.25, 12.5, and 25 μM) for 24–72 h by MTT assay. Our experiment found that high-dose of SFN could decrease cell proliferation. To demonstrate that SFN reduces proliferation whether the cause of apoptosis/necrosis by annexin V-FITC assay. The data showed that SFN treated cells had nonsignificant increase in the percentage of apoptotic cells com-pared with the untreated (control) cells. Thus, we hypothesized that the proliferation of these cells was inhibited by pathways other than those of apoptosis/necrosis.

(As Result; Lines 211-212)

Comment 6: In line 243-247, although the authors described that SFN-induced growth arrest occurred by mechanisms other than intracellular ROS accumulation, the results in Fig. S1C appear to cause ROS accumulation due to SFN. Please describe the basis for this claim in the discussion section.

Response to the comment 6:

Agree. Thanks to the reviewers for pointing out this problem. We describe the SFN-induced growth arrest and ROS accumulation. We found that SFN reduced ROS accumulation in MDA-MB-231 and ZR-75-1 cells, suggesting that SFN-induced growth arrest in MDA-MB-231 cells was caused by inhibition of intracellular ROS accumulation leading to cell growth inhibition.

(As Result; Lines 255-258)

Comment 7: In line 277-279, please describe the results of the ZR-75-1 cells.

Response to the comment 7:

Thanks to the reviewers for pointing out this problem. Our results found MPM-2 levels in ZR-75-1 cells treated with SFN to be higher than those in the control group. MPM-2 (anti-phospho-Ser/Thr-Pro)-FITC (green in Figure 2A), an antibody able to rec-ognize proteins with epitopes exclusively phosphorylated during mitosis, specifically from the early prophase to metaphase.

(As Result 3.5.; Lines 277-279)

Comment 8: In line 300-301, please describe the results of qPCR analyses.

Response to the comment 8:

Thanks to the reviewers for pointing out this problem. Gene expression analysis showed that cyclin B1 upregulated, but CDC2 levels not significantly changed following incubation with SFN in MDA-MB-231 and ZR-75-1 cells.

(As Result 3.6.; Lines 298-300)

Comment 9: In line 313-315, please describe the results of the ZR-75-1 cells.

Response to the comment 9:

Thanks to the reviewers for pointing out this problem. Western blot analysis revealed a increase in the CDC25C: p-CDC25C ratio in ZR-75-1 cells after incubation with SFN. These data suggest that SFN is involved CDC25C regulation in MDA-MB-231 and ZR-75-1 cells.

(As Result 3.7.; Lines 315-318)

Comment 10: In line 405-406, in order to apply it as anticancer agent, it is necessary to consider the effect on normal cells. Please cite some papers and describe the effect on normal cells in the discussion section.

Response to the comment 10:

Thanks to the reviewers for pointing out this problem. We agree with this comment. We describe the absence of significant observed SFN inhibition of human normal cell. It shows that SFN has specific poison to kill cancer cells.

(As Conclusions; Lines 405-410)

References:

Chen, Xin, et al. "Activation of Nrf2 by sulforaphane inhibits high glucose-induced progression of pancreatic cancer via AMPK dependent signaling." Cellular Physiology and Biochemistry 50.3 (2018): 1201-1215.

Comment 11: Some experiments have shown different results for the two cell lines. Please discuss in the discussion section the pathway through which SFN-induced growth arrest of ZR-75-1 cells occurs.

Response to the comment 11:

Thanks to the reviewers for pointing out this problem. We agree with this comment. Our results provide experimental evidence that SFN inhibits MDA-MB-231 and ZR-75-1 cell growth. We observed that treating cancer cells with SFN resulted in the downregulation of cyclin B1/CDC2 complex association as well as CDC25C phosphorylation in MB-231 and ZR-75-1 cells. Our experimental evidence thus indicates that SFN may irreversibly arrest the growth of cancer cells and that the inhibition of proliferation and induction of cell cycle arrest are both highly dependent on the of SFN accumulation in cancer cells.

(As Discussion; Lines 381-404)

Reviewer 3 Report

Query#1

First, in the abstract is stated that the dissociation of cyclin B1/CDC2 induced by SFN may also determine antitumor activity in breast cancer. Please clarify this sentence and explain why the dissociation of cyclin B1/CDC2 could predict antitumor activity in breast cancer. Surely this process is well explained in the rest of the article; indeed, is presented a special paragraph entitled: “3.6. G2/M-phase cell cycle arrest by SFN in the MDA-MB-231 cells through CDC2 and cyclin B1 disassociation”, in which the authors clearly explain this process, however being the abstract the paper’s calling card, in my opinion it is better to describe more clearly what they meant.

Query#2

In the introduction I would suggest to the authors to stress the importance of the natural microenvironment as a huge resource of novel lead compounds, belonging to a great variety of different chemical structural classes containing different heterocycle ring and different heteroatoms.

At this purpose I suggest to the authors to cite the following papers:

1)     Hashem, S., Ali, T. A., Akhtar, S., Nisar, S., Sageena, G., Ali, S., Al-Mannai, S., Therachiyil, L., Mir, R., Elfaki, I., Mir, M. M., Jamal, F., Masoodi, T., Uddin, S., Singh, M., Haris, M., Macha, M., & Bhat, A. A. (2022). Targeting cancer signaling pathways by natural products: Exploring promising anti-cancer agents. Biomedicine & pharmacotherapy = Biomedecine & pharmacotherapie150, 113054. https://doi.org/10.1016/j.biopha.2022.113054

2)     Carbone, D., Vestuto, V., Ferraro, M. R., Ciaglia, T., Pecoraro, C., Sommella, E., Cascioferro, S., Salviati, E., Novi, S., Tecce, M. F., Amodio, G., Iraci, N., Cirrincione, G., Campiglia, P., Diana, P., Bertamino, A., Parrino, B., & Ostacolo, C. (2022). Metabolomics-assisted discovery of a new anticancer GLS-1 inhibitor chemotype from a nortopsentin-inspired library: From phenotype screening to target identification. European journal of medicinal chemistry, 234, 114233. https://doi.org/10.1016/j.ejmech.2022.114233

Moreover, considering the activity of SFN on CDKs, I suggest to the authors to emphasize that other semi-synthetic and synthetic derivatives inspired from natural molecules have been reported in literature demonstrating antitumor activity against several types of cancer, with ability to hamper enzymatic activity of protein kinases.

At this purpose I suggest to cite the following updated literature:

1.     Baier, A., & Szyszka, R. (2020). Compounds from Natural Sources as Protein Kinase Inhibitors. Biomolecules, 10(11), 1546. https://doi.org/10.3390/biom10111546

2.     Pecoraro, C., Parrino, B., Cascioferro, S., Puerta, A., Avan, A., Peters, G. J., Diana, P., Giovannetti, E., & Carbone, D. (2021). A New Oxadiazole-Based Topsentin Derivative Modulates Cyclin-Dependent Kinase 1 Expression and Exerts Cytotoxic Effects on Pancreatic Cancer Cells. Molecules (Basel, Switzerland), 27(1), 19. https://doi.org/10.3390/molecules27010019

Query#3

Paragraph 3.5. Effects of SFN on the mitotic index. Is not clear the effect of SFN on mitotic index. Please clarify this point.

Query#4

 In general, the paper is well organized and supported by the experimental data obtained, however I would suggest the authors to include in the introduction that through this piece of paper they try to explain the mechanism by which SFN induces reduction of cell growth in breast cancer cells. As, from the introduction it would seem as if from the beginning the targets are CDK5 and the complex cyclinB1/CDC2. In truth, it is the experimental evidence that leads the authors to hypothesize the target.

Author Response

Department of Surgery

Kaohsiung Municipal Ta-Tung Hospital

Kaohsiung, Taiwan

February 27th, 2023

Dear Editor:

Thank you for giving me the opportunity to submit a revised draft of our manuscript which titled: Sulforaphane-induced cell mitotic delay and-inhibited cell proliferation via regulating CDK5R1 upregulation in Breast cancer cell lines. We appreciate the time and effort that you and the reviewers have dedicated to providing your valuable feedback on our manuscript. We are grateful to the reviewers for their insightful comments on this paper. We have been able to incorporate changes to reflect most of the suggestions provided by the reviewers. We have highlighted the modification within the manuscript and point-by-point response to the reviewers’ comments and concerns in this revision.

Comments from Reviewer 3:

Comment 1: First, in the abstract is stated that the dissociation of cyclin B1/CDC2 induced by SFN may also determine antitumor activity in breast cancer. Please clarify this sentence and explain why the dissociation of cyclin B1/CDC2 could predict antitumor activity in breast cancer. Surely this process is well explained in the rest of the article; indeed, is presented a special paragraph entitled: “3.6. G2/M-phase cell cycle arrest by SFN in the MDA-MB-231 cells through CDC2 and cyclin B1 disassociation”, in which the authors clearly explain this process, however being the abstract the paper’s calling card, in my opinion it is better to describe more clearly what they meant.

Response to the comment 1:

Agree. Thanks to the reviewers for pointing out this problem. We describe more clearly SFN inhibits MDA-MB-231 and ZR-75-1 cell growth. We observed that treating cancer cells with SFN resulted in the downregulation of cyclin B1/CDC2 complex association as well as CDC25C phosphorylation in MB-231 and ZR-75-1 cells. Our experimental evidence thus indicates that SFN may irreversibly arrest the growth of cancer cells and that the inhibition of proliferation and induction of cell cycle arrest are both highly dependent on the of SFN accumulation in cancer cells.

Comment 2: In the introduction I would suggest to the authors to stress the importance of the natural microenvironment as a huge resource of novel lead compounds, belonging to a great variety of different chemical structural classes containing different heterocycle ring and different heteroatoms.

At this purpose I suggest to the authors to cite the following papers:

1)     Hashem, S., Ali, T. A., Akhtar, S., Nisar, S., Sageena, G., Ali, S., Al-Mannai, S., Therachiyil, L., Mir, R., Elfaki, I., Mir, M. M., Jamal, F., Masoodi, T., Uddin, S., Singh, M., Haris, M., Macha, M., & Bhat, A. A. (2022). Targeting cancer signaling pathways by natural products: Exploring promising anti-cancer agents. Biomedicine & pharmacotherapy = Biomedecine & pharmacotherapie, 150, 113054. https://doi.org/10.1016/j.biopha.2022.113054

2)     Carbone, D., Vestuto, V., Ferraro, M. R., Ciaglia, T., Pecoraro, C., Sommella, E., Cascioferro, S., Salviati, E., Novi, S., Tecce, M. F., Amodio, G., Iraci, N., Cirrincione, G., Campiglia, P., Diana, P., Bertamino, A., Parrino, B., & Ostacolo, C. (2022). Metabolomics-assisted discovery of a new anticancer GLS-1 inhibitor chemotype from a nortopsentin-inspired library: From phenotype screening to target identification. European journal of medicinal chemistry, 234, 114233. https://doi.org/10.1016/j.ejmech.2022.114233

Moreover, considering the activity of SFN on CDKs, I suggest to the authors to emphasize that other semi-synthetic and synthetic derivatives inspired from natural molecules have been reported in literature demonstrating antitumor activity against several types of cancer, with ability to hamper enzymatic activity of protein kinases.

At this purpose I suggest to cite the following updated literature:

  1. Baier, A., & Szyszka, R. (2020). Compounds from Natural Sources as Protein Kinase Inhibitors. Biomolecules, 10(11), 1546. https://doi.org/10.3390/biom10111546
  2. Pecoraro, C., Parrino, B., Cascioferro, S., Puerta, A., Avan, A., Peters, G. J., Diana, P., Giovannetti, E., & Carbone, D. (2021). A New Oxadiazole-Based Topsentin Derivative Modulates Cyclin-Dependent Kinase 1 Expression and Exerts Cytotoxic Effects on Pancreatic Cancer Cells. Molecules (Basel, Switzerland), 27(1), 19. https://doi.org/10.3390/molecules27010019

Response to the comment 2:

Agree. Thanks to the reviewers for pointing out this problem. We modified and updated the new references to support our manuscript. Natural products have been an unparalleled source of anti-cancer drugs in the modern age of drug discovery. Natural products play an important role in cancer therapy by modulating the cancer microenvironment and different signaling pathways. Therefore, it is necessary to study the current role of natural products in the discovery and development of anti-cancer drugs and determine the possibility of natural products as an important source of future therapeutic agents.

(As References; 6 and 12)

Comment 3: Paragraph 3.5. Effects of SFN on the mitotic index. Is not clear the effect of SFN on mitotic index. Please clarify this point.

Response to the comment 3:

Thanks to the reviewers for pointing out this problem. We describe the effect of SFN on mitotic index. We found MPM-2 levels in MDA-MB-231 and ZR-75-1 cells treated with SFN to be higher than those in the control group. However, the effects of staining with MPM-2 were as strong as those of nocodazole in the group treated with 25 μM SFN, likely because cells stained with MPM-2 were in various stages of mitosis, some of which could not be identified using this early pro-phase marker. In particular, some G2/M phase cells may have remained unstained. Thus, in addition to suggesting mitotic disturbance, increased MPM-2 staining may suggest some underestimation the degree of mitotic disturbance. Taken together, it is shown that MDA-MB-231 and ZR-75-1 cells were treated with SFN to increase MPM-2 levels, which resulted in G2 arrest from mitotic arrest.

(As Result 3.5.; Lines 278-293)

Comment 4: In general, the paper is well organized and supported by the experimental data obtained, however I would suggest the authors to include in the introduction that through this piece of paper they try to explain the mechanism by which SFN induces reduction of cell growth in breast cancer cells. As, from the introduction it would seem as if from the beginning the targets are CDK5 and the complex cyclinB1/CDC2. In truth, it is the experimental evidence that leads the authors to hypothesize the target.

Response to the comment 4:

Agree. Thanks to the reviewers for pointing out this problem. We added to the introduction that SFN induces a reduction in breast cancer cell growth to support our manuscript.

(As Background; Lines 57-58)

Additional clarifications

In addition to the above comments, all spelling and grammatical errors pointed out by the reviewers have been corrected.

Sincerely,

Dr. Kuan-Ting Lee

Department of Surgery, Kaohsiung Municipal Ta-Tung Hospital, Kaohsiung, Taiwan

E-mail: ayta860404@gmail.com

Round 2

Reviewer 2 Report

My comments have been addressed adequately.

Please check out the font of the words below:

In lines 24, 170 and 345, are "cdc25c" and "CDC25C" used separately?

In line 106, please change the "2" in "CO2" to a subscript.

In lines 281, 296, 371 and 403, please change the "2" in "G2" to a subscript.